# The Impacts of Urban Form on Carbon Emissions: A Comprehensive Review

Changlong Sun [1,†], Yongli Zhang [2,†], Wenwen Ma [1], Rong Wu [2] and Shaojian Wang [3,*]

1   School of Economics and Management, Wenzhou University of Technology, Wenzhou 325000, China
2   School of Architecture and Urban Planning, Guangdong University of Technology, Guangzhou 510062, China
3   School of Geography and Planning, Sun Yat-sen University, Guangzhou 510275, China
*   Correspondence: wangshj8@mail.sysu.edu.cn
†   These authors contributed equally to this work.

**Abstract:** As a result of global climate change and urban development, the interaction between urban form and carbon emissions has become a frontier issue and a key area of carbon emission research. This paper presents a scientometric analysis of 2439 academic publications between 2002 and 2021 on urban form and carbon emissions to explore the current state of global research and future development potential. Citespace and VOSviewer were the primary analysis tools. The results showed the following: (1) The number of articles published on urban form and carbon emission research shows an increasing trend, especially after 2012. (2) Scientific research institutions and authors in developed countries paid attention sooner to the urban ecological environment. With the deepening of economic globalization, developing countries began to pay more attention to the urban environment. (3) Through an analysis of keyword clusters, timelines, and stacked area charts, the development of the urban form and carbon emissions can be divided into the following three stages. The first is the budding stage, which is characterized by preliminary research on the atmospheric environmental impact factors. The second stage is the development stage, with urban areas becoming the leading research object of carbon theory. The third stage is the mature stage, which is characterized by an emphasis on the optimization of carbon emissions. (4) Finally, the influence of urban form on carbon emissions includes four main aspects: land use, built environment, transportation networks, and development patterns.

**Keywords:** carbon emission; urban form; correlation; knowledge map

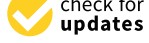

## 1. Introduction

Since the end of the 20th century, carbon emissions have received significant attention in academic circles due to global warming and climate change. Recently, carbon emissions were emphasized in the 2021 Intergovernmental Panel on Climate Change (IPCC) Sixth Assessment Report (AR6), which listed "Human influence warms the atmosphere, oceans, and land" as the first title. The report pointed out that the concentration of carbon dioxide in the atmosphere (410 ppm) is the highest it has been in 2 million years. The global average temperature will rise by at least 1.5 °C by the middle of the 21st century due to anthropogenic emissions [1], which poses a significant threat to the survival of human beings and the sustainable development of the environment. In addition, 55% of the world's population lived in cities as of 2018, and the global urbanization rate is expected to reach 68% by 2050 [2]. Cities consume approximately 75% of the world's resources and generate more than 70% of greenhouse gases [3]; thus, cities are the main places where carbon emissions are generated. Urban form can be defined as a process with temporal and spatial boundaries, including the spatial organization and arrangement of human activities, and, at the same time, can affect the development and expansion of the city and the efficiency of resource allocation, land use, transportation, and infrastructure, and better promote city

development [4,5]. Diverse research results on urban form and carbon emissions have been obtained by the authors from different disciplinary backgrounds and different countries. Most authors believe that urban forms play an essential role in improving urban land-use efficiency [6], compactness [7], population density [8], and energy efficiency [9]. Among these, land use [10], energy efficiency [11,12], and other factors are closely related to changes in carbon emissions. Although there are many studies on the correlation between urban form and carbon emissions, the previous review literature expounds the concept of urban form and the correlation between energy consumption, but few studies have discussed the influencing factors of urban form and carbon emissions [13], and some reviews have only discussed the impact of urban form on transportation carbon emissions [14]. This paper not only considers the carbon emissions of transportation, but also the impact of urban morphological factors such as the built environment and three-dimensional urban structure on the carbon emissions of buildings. It also summarizes some research methods commonly used. Thus, it is more systematic and comprehensive than previous reviews. In addition, some difficulties in previous studies have not been solved effectively: (1) Firstly, due to the diversity of indicators, the quantitative research on the impact of urban form on carbon emissions still needs to be further improved. (2) In addition, the impact of urban form on carbon emissions at different stages of development and different types of cities is unclear. (3) Finally, low-carbon urban development strategies still need to be enriched (e.g., the optimization of the land use, built environment, traffic network, and development pattern). These problems need to be solved urgently to improve the urban environment and achieve sustainable urban development, meanwhile, it will clarify the correlation between carbon emissions and urban form. Therefore, it is important to review previous studies and carry out a systematic bibliometric analysis to understand the future development directions.

## 2. Materials and Methods

### 2.1. Data

This study used the Web of Science (WOS) database as the literature data source to represent the global research situation. The WOS was selected, and the literature retrieved covered from 2002 to 2021 (data as of December 2021). The topics of the search were input as "#1:TS = urban spatial structure, urban structure, and urban form" and "#2:TS = carbon emissions and carbon neutral." Article and review literature types were selected to obtain a total of 2439 retrieved results. Finally, the obtained literature data were exported in txt format.

### 2.2. Methods

Visualization tools can enable domain analysis for science and technology management by visually presenting data in graphs [15]. Moreover, they can provide multivariate and dynamic visualization analysis of data. Among these tools, Citespace and VOSviewer (VOS) are used widely [16,17]. To date, these tools have been used in the visual analysis of various disciplines, such as infectious disease prediction [18], medical data mining [19], global food security [20], urban resilience assessment [21], the development history of smart cities [22], and geographic information system (GIS) knowledge fields [23]. In this study, VOS and Citespace were the primary tools to analyze and visualize the core contents with their powerful literature analysis functions. Furthermore, based on bibliometric analysis, a reading of highly cited literature was carried out to assess and summarize the main research content and trends related to the current global urban form and carbon emissions.

## 3. Results

### 3.1. Overall Results

Figure 1 shows the evolution of published papers on urban form and carbon emissions during the 20 years of 2002–2021. Over time, the growing tendency indicates that more and more scientific research on urban form and carbon emissions has been conducted, and two stages can be identified as follows:

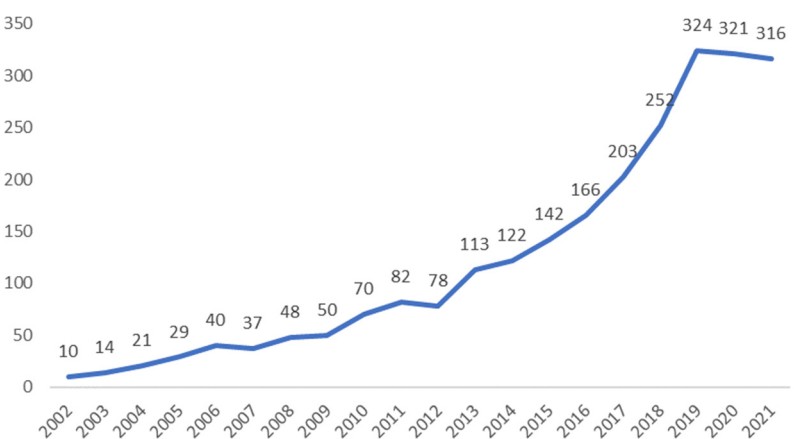

**Figure 1.** The number of global publications on urban form and carbon emissions (2002–2021).

3.1.1. Development Phase (2002–2012)

During this stage, global researchers produced books, reports, and many empirical types of research on urban form and carbon emissions. Each year, 5 to 10 additional articles were published, indicating that urban form became an essential factor affecting carbon emissions [24–27]. In addition, the Kyoto Protocol came into force in 2005, and greenhouse gas emissions were regulated for the first time in human history, contributing to the growth in research output.

3.1.2. Rapid Development Phase (2013–2021)

Since 2013, the number of articles on urban form and emissions increased promptly, and more than 1959 articles published during this period. The World Climate Change Conference organized in Doha in 2012 emphasized the importance of implementing the Kyoto Protocol II commitments. Moreover, scholars from all over the world began to pay attention to the influence mechanism of urban form on carbon emissions at different scales and the possibility of using the characteristics of urban form to reduce carbon emissions [28–32].

Figures 2 and 3 show that the developed countries in Europe and the United States had the highest number of papers published in the early stage. In recent years, the research of emerging economies such as China and India has increased rapidly. A comparative analysis shows that developed countries in Europe and the United States have a relatively high level of urbanization and have made progress in estimating and influencing carbon emissions related to urban form. However, the urbanization levels of countries in Asia, South America, and Africa are increasing rapidly, and the contradiction between the development of urban form and carbon emissions is becoming increasingly prominent.

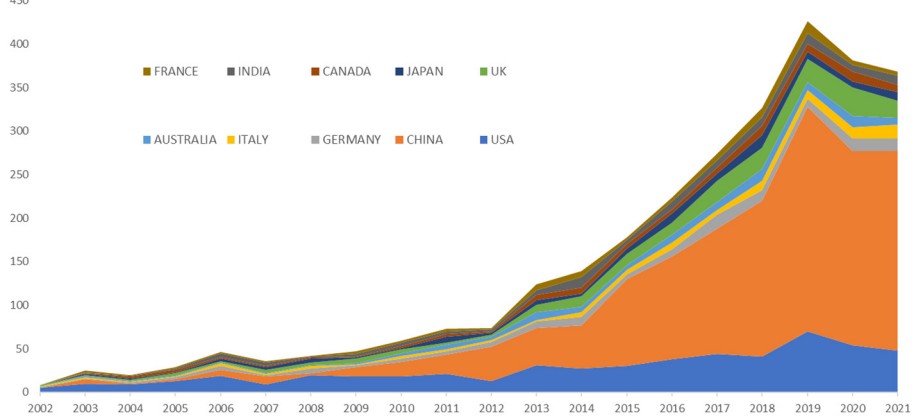

**Figure 2.** Stacked area charts of global publications (Top 10).

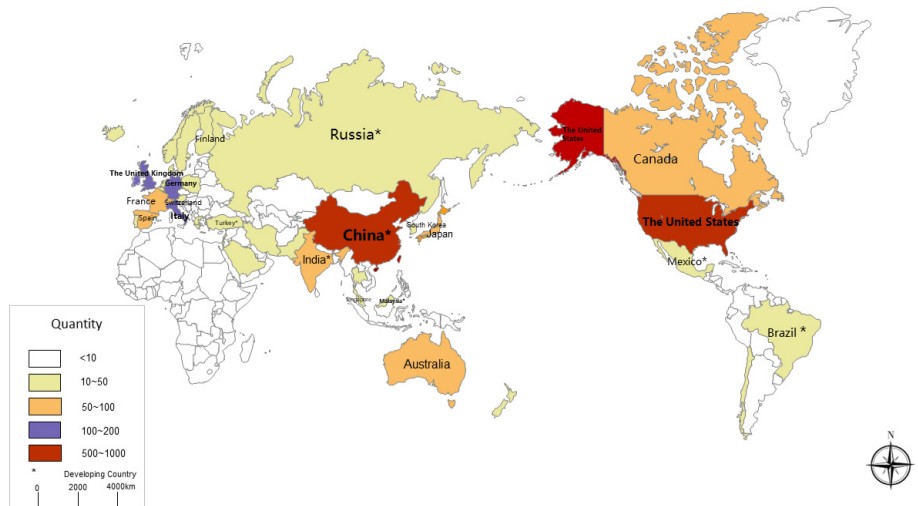

**Figure 3.** Spatial distribution of global publications.

The distribution of journals shows that the *Journal of Cleaner Production* has the highest number of articles (160), followed by *Atmospheric Environment and Sustainability* (109), and the other journals account for less than 100 articles. In addition, when classifying the subjects of the WOS database (Table 1), it was found that the research is concentrated in the fields of ecological and environmental science (81.87%), energy and fuel (42.63%), meteorology and atmospheric science (41.41%), and engineering (38.68%); because a document may belong to multiple categories, the total is more than 100%. The distribution of subject categories suggests the field's ecological, geographical, atmospheric, and environmental issues were prioritized. Moreover, urban form and carbon emission research became more interdisciplinary over time.

**Table 1.** Subject classification of the Web of Science.

| Subject Classification | Frequency | Proportion |
|---|---|---|
| Environmental Sciences Ecology | 1888 | 81.87% |
| Energy Fuels | 983 | 42.63% |
| Meteorology Atmospheric Sciences | 955 | 41.41% |
| Engineering | 892 | 38.68% |
| Business Economics | 657 | 28.49% |
| Science Technology Other Topics | 650 | 28.18% |
| Geography | 611 | 26.49% |
| Public Environmental Occupational Health | 575 | 24.94% |
| Physical Sciences Other Topics | 475 | 20.59% |
| Mathematics | 293 | 12.71% |

*3.2. Keywords Co-Occurrence Network*

In this study, VOSviewer was used to analyze author keywords in the data, and the occurrence frequency threshold was set to 5. In addition, the result generated a total of 220 keywords and the keyword cluster atlas was drawn (Figure 4). The keywords are divided into 12 clusters, and we mainly consider those associated with the carbon emissions of urban settlement areas, because urban settlements make a large contribution to carbon emissions. The results in Figures 4 and 5 suggest that there are five core research topics that are closely relevant: carbon emission, urban form, sustainable, land use, and energy efficiency.

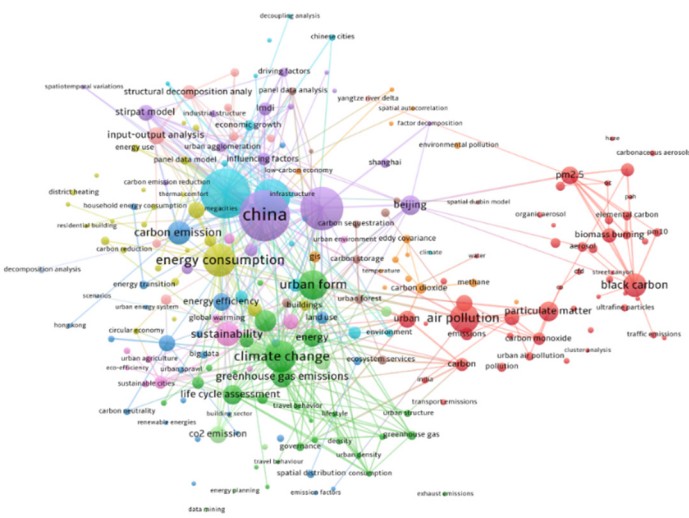

**Figure 4.** Keywords co-occurrence network.

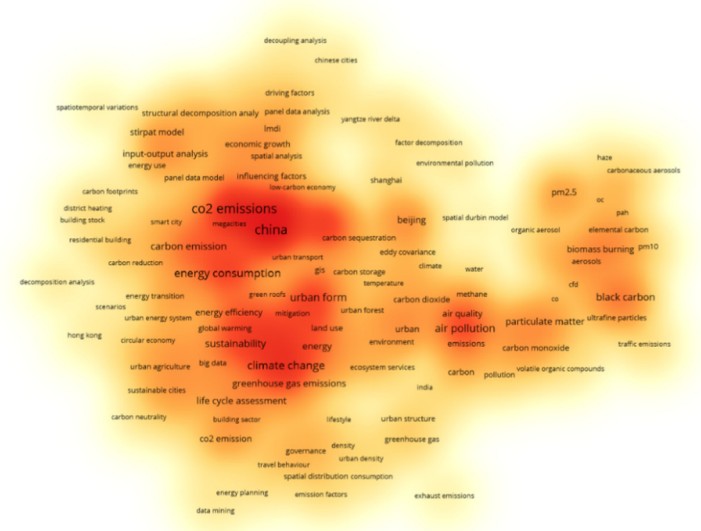

**Figure 5.** Keywords hotspot map.

Citespace was used for keyword emergence analysis, which found that the keywords with a high burst value related to the topic of this paper are carbon element, city, and particle. Furthermore, their emergence values are all over 10, indicating that these keywords have become research hotspots over a certain period (Figure 6). Overall, three stages of emergence from 2002 to 2021 were identified. The nascent stage appeared around 2002. In this stage, carbon element and atmospheric environment were the main research topics [33,34]. The development stage appeared around 2004, during which city became an essential direction of carbon emission research, and research on urban form and carbon emission continued to flourish. The mature stage appeared around 2019. The main keywords in this stage were Carbon dioxide ($CO_2$) emissions, empirical analysis, and optimization, indicating that the research on urban form and carbon emissions gradually matured, and focused on the influence factors such as built environment, traffic, and land use.

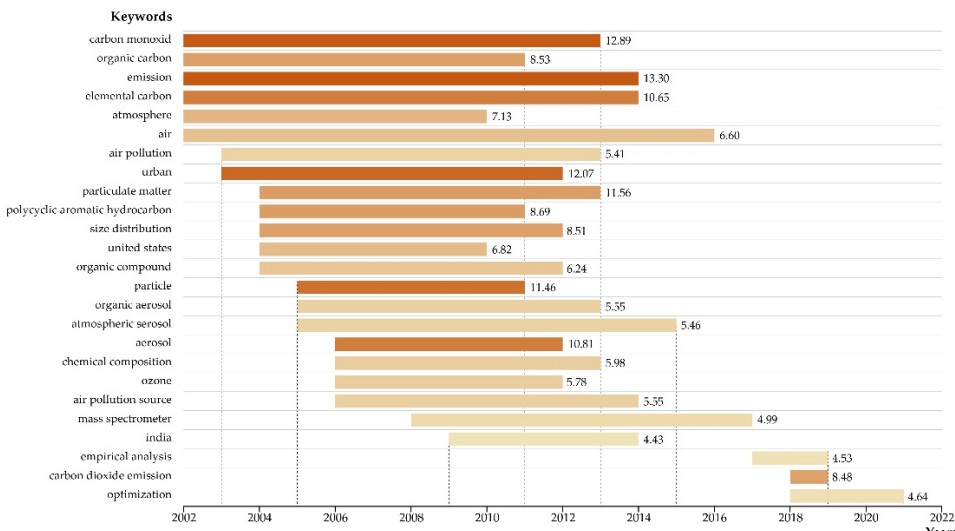

**Figure 6.** The top 25 keywords with the strongest citation burst words for the WOS. The data for this figure come from Citespace.

### 3.3. Document Co-Citation Network

The Figure 7 shows the document co-citation network (DCN), which consists of 1041 cited references and 4742 co-citation links between 2002 and 2021. The silhouette scores of the most crucial 10 clusters are all above 0.8, which mean the result of the figure are credible (Table 2). The largest cluster—#0 Secondary organic aerosol formation and #1 Urban form—contains 133 member references, and is thus slightly more extensive than the others. In conclusion, with the change of the atmospheric environment, urban form and city size have become the main focus in the research field of urban carbon emissions.

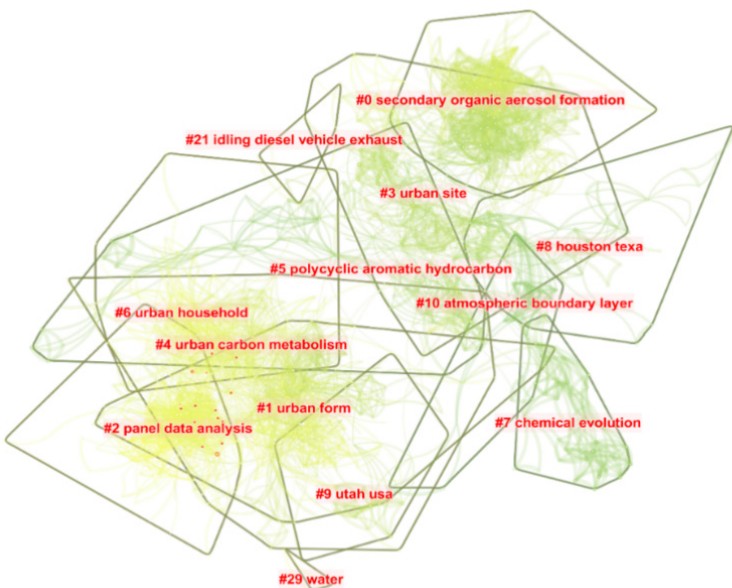

**Figure 7.** The document co-citation network. The "#" and number indicate the serial number of the clusters with the largest to the smallest number.

**Table 2.** Summary of the largest 10 clusters.

| Cluster ID | Size | Silhouette Score | Label (LLR) | Mean (Cite Year) |
|---|---|---|---|---|
| 0 | 133 | 0.925 | Secondary organic aerosol formation | 2006 |
| 1 | 133 | 0.893 | Urban form | 2007 |
| 2 | 125 | 0.902 | Panel data analysis | 2011 |
| 3 | 109 | 0.875 | Urban site | 2002 |
| 4 | 97 | 0.89 | Urban carbon metabolism | 2009 |
| 5 | 83 | 0.919 | Polycyclic aromatic hydrocarbon | 1996 |
| 6 | 68 | 0.92 | Urban household | 2010 |
| 7 | 35 | 0.965 | Chemical evolution | 2001 |
| 8 | 34 | 0.97 | Houston, Texas | 2001 |
| 9 | 22 | 0.982 | Utah, USA | 2002 |

Table 3 shows the ten papers with a total cited number of more than 46. Among these ten papers, nine belong to clusters #1, #2, and #4, indicating that the research perspective in these papers mainly focuses on the impact of urban form on carbon emissions. Panel data analysis is a popular method used to explore the relationship between urban form and carbon emissions. According to the statistics of authors with high co-citations, Dhakal ranks first with 73 co-citations. He analyzed urban carbon emissions at the national, meso-, and micro-scales in 2009 and found that cities accounted for 84% of China's commercial energy use. Thirty-five significant cities in China account for 18% of the total population; however, they account for 40% of China's energy consumption and $CO_2$ emissions, which provides a scientific basis for future research on urban carbon emissions [35].

**Table 3.** The top 10 co-citation frequencies of the papers.

| Citation Counts | References | Cluster ID |
|---|---|---|
| 73 | Dhakal S,2009, ENERG POLICY | 4 |
| 67 | Glaeser EL,2010, J.ECOL. | 1 |
| 52 | Ewing R,2008, HOUS POLICY DEBATE | 1 |
| 51 | York R,2003, ECOL ECON | 2 |
| 51 | Poumanyvong P,2010, ECOL ECON | 2 |
| 51 | Fang CL,2015, APPL ENERG | 2 |
| 49 | Ewing R,2010, J Am Plann Assoc | 1 |
| 48 | Wang SJ,2017, APPL ENERG | 2 |
| 47 | Mi ZF,2016, APPL ENERG | 4 |
| 46 | Ang BW,2004, ENERG POLICY | 6 |

*3.4. Research Methods*

3.4.1. Accounting Methods of Carbon Emission

In the process of reading the literature in this paper, we found that some scholars believe that the accounting boundary of carbon emissions is mainly classified according to the production side, the consumption side, and the "emission raising" [36]. Other scholars have proposed more specific accounting boundaries, namely, the geographic boundary, the temporal boundary, the activity boundary, and the life cycle boundary [37]. These boundary theories are the basis of the carbon emission accounting method, in which we take into account the differences in the scale of urban form research, and the urban carbon emission accounting methods are divided into the direct method, indirect method, and measurement method.

The direct method is suitable for accounting for carbon emissions in large cities or cities with a high degree of development. The principle of the direct method is based on the emission inventory provided by IPCC [38]. The carbon emission factors and conversion coefficients of different types of energy are multiplied by a city's energy consumption to obtain the corresponding carbon emissions. The use of the direct method depends on the

availability and integrity of energy data. At the same time, large or highly developed cities have relatively complete data statistical systems and high data openness, which facilitates the use of the direct method [39–43].

The indirect method is generally used in small and medium-sized cities or less-developed regions. The indirect method is mainly based on proxy data (e.g., nighttime light (NTL) and land use), transforming carbon emission data into large spatial units and distributing them onto a spatial grid for estimation [44]. With the development of GIS and remote sensing (RS) in recent years, NTL data have become the primary proxy data in regional carbon emission accounting [45,46]. The Defense Meteorological Satellite Program's Operational Line-scan System (DMSP-OLS) introduced in 1992 provides the most widely used NTL data [47–49]. Studies have shown that economic development positively correlates with carbon emissions.

In contrast, economic development positively correlates with the level of urbanization. Further, the level of urbanization is positively correlated with a city's stable light intensity, so that NTL can well represent a city's carbon emissions level. At the same time, the availability of energy consumption data in small and medium-sized cities is low, and the direct method is challenging to be used. Therefore, as an important source of proxy data of the indirect method, NTL is suitable for accounting for carbon emissions in small and medium-sized cities.

It is suitable to adopt an actual measurement method when conducting small-scale carbon emission accounting, such as in communities or parts of cities. The measurement method mainly allocates emissions to designated spatial units by collecting and organizing data at the point or sector level [50,51]. Researchers in the UK used low-cost portable $CO_2$ sensors to monitor and estimate $CO_2$ dispersion in the old city of Edinburgh [52]. Zhang et al. (2018) attempted to link greenhouse gas emissions measured at the sectoral level with spatial land use, enabling spatial visualization at different scales such as districts or streets [53]. Compared with the previous two methods, the carbon emission data obtained by the measurement method is more accurate. However, it requires more workforce and material resources, so selecting the method should align with practical needs.

According to the current status of the research literature, it is obvious that these three methods provide particular convenience for accounting for urban-space carbon emissions. However, there are still some problems with the current accounting methods. Although the direct method is convenient for accounting, it has a limited scope of application and a high degree of dependence on energy consumption data; it needs to be implemented with a corresponding and accurate statistical system, and there is still controversy over the fairness of carbon accounting for energy-producing cities. Notwithstanding the fact that the indirect method improves the calculation efficiency, it is limited by low spatial resolution and oversaturation of solid light in urban areas. Furthermore, although the estimation results of the measured method are more accurate than those of the first two methods, this accuracy is dependent on local relevant data and the cost of data collection is high [54]. Therefore, improving the applicability and optimization of accounting methods is the main goal for the future.

### 3.4.2. Methods to Explore the Correlation between Urban Form and Carbon Emissions

The current research methods of urban form and carbon emissions can be mainly divided into spatial methods and non-spatial methods. Spatial methods mainly use geographic information or spatial distribution data to explore the spatial differentiation and spillover effect or the impact mechanism of carbon emissions, and include geographic weighting models, spatial autocorrelation models, and GIS spatial analysis [55–57]. For example, Wang et al. used a spatial autocorrelation model to explore the spatial spillover effect and driving force of China's urban carbon emissions [58]. Furthermore, non-spatial types of data (e.g., economic status, population density, and energy consumption) is frequently used to identify the impact of different factors of urban form on carbon emissions in the non-spatial methods, which include linear or the nonlinear correlation models, panel

data models, and index decomposition methods (e.g., the Kaya identity model and STIRPAT model) [53,54,59,60]. For example, Yang et al. found that China's urban form and traffic factors have a significant impact on per capita carbon dioxide emissions through a panel data model [61]. Van et al. used a multiple linear regression model to study the impact of the built environment on residents' travel carbon emissions, and found that the average travel time was negatively correlated with rail transit density [62].

With the improvement in data acquisition capabilities and the development of technology, new methods are constantly emerging. To avoid the negative consequences of multicollinearity between explanatory variables and improve the accuracy of the results, the random forest regression model based on machine learning is widely applied in research. Lin et al. found through a random forest regression model that the degree of spatial crowding plays a dominant role in the variation of $CO_2$ emissions [63]. In short, there are many types of research methods for examining the correlation between urban form and carbon emissions, which need to be appropriately selected according to different needs to ensure the validity of the results.

*3.5. Impacts of Urban Form on Carbon Emissions*

As a social and technological system, urban form significantly influences $CO_2$ emissions [64,65]. This paper finds that the study of urban form and carbon emissions can be roughly divided into two types of literature: (1) analyses of the influencing factors of $CO_2$ emissions through quantitative research on land- use change, built environment, transportation, and a development model of the urban form; and (2) proposing new methods and strategies to accelerate the development of low-carbon cities by exploring the mechanism of different influencing factors of urban form.

3.5.1. Land Use

As the spatial carrier of urban activities, the land is an essential manifestation of urban spatial structure and form. Land-use research mainly considers type, scale, efficiency, structure, and management. According to land-use patterns and industrial types, land-use types are divided into agricultural land, industrial land, commercial and residential land, transportation land, and other land [66]. Once the land-use type changes, it directly affects carbon sequestration in terrestrial ecosystems and indirectly affects anthropogenic carbon emissions [67–69]. It is generally believed that the carbon sink will decrease with the transformation of agricultural land (forest land, grassland, farmland) to construction land (industrial land, commercial land, residential land, transportation land) [70,71]. Furthermore, urban expansion will lead to fragmentation and irregularity of various land-use patches in cities, which will make the urban form highly complex, and economic activities will be scattered into different pieces, resulting in more carbon emissions [72–76]. In addition, the level of land-use efficiency, the benefits and drawbacks of structure, and the implementation of land-management policies will also have varying degrees of impact on urban carbon emissions [77–79]. Huang et al. (2013) explored the scale and spatial distribution of various types of land by building a regression model and found that the concentrated distribution of industrial land may reduce the carbon emission intensity [80]. However, the concentrated distribution of the other four land-use types will promote carbon emissions.

With the deepening of research, land-use structure and land-management policies are considered the main land-use influencing factors affecting carbon emissions [81]. Consequently, the emission reduction strategy of land use should optimize the land-use structure and land management. There are differences in the ability of different types of land to generate and absorb $CO_2$. Their functional attributes and structures significantly affect urban residents' travel mode and energy consumption. In order to optimize the land-use structure, the first step is to promote high-carbon source land to high-carbon land change. Researchers have explored the carbon sinks of urban parks in different countries and found that urban parks can offset 2.3 to 3.6% of their annual greenhouse gas emissions [82–84].

Therefore, increasing the proportion of carbon sinks such as parks and green spaces in cities will help reduce the pressure of emissions there. The next step is to strengthen the mixed use of land. Studies have suggested that mixed-use land can reduce the travel distance and shorten the travel time of residents, thereby changing the way residents travel to achieve the effect of emission reduction [85–89]. The mixed use of land functions includes horizontal and vertical two-dimensional mixing. Horizontal mixing refers to low-density areas with residential, commercial, and green spaces, which are usually used on a small scale to create a comfortable small environment, while vertical mixing refers to the vertical space of high-rise buildings. As an example of a mixed-use high-density area, the lower level of a building could be the commercial area, the middle level an office area, and the upper level a living area. Additionally, it is also crucial to formulate a sound land-management system, improve the development efficiency of other types of land, spatially achieve sustainable and balanced development, and avoid the imbalance of land-use structure to reduce the burden of carbon emissions. In summary, in-depth quantitative research should be continued in the future to explore better solutions for land use and to build urban land-use patterns and urban forms that are comfortable and sustainable for living.

### 3.5.2. Built Environment

In terms of the built environment, numerous indicators have been used by different researchers in recent years. The 3D indicators (Density, Diversity, and Design) proposed by Ewing and Kockelman are widely accepted [90]. Subsequently, other indicators have emerged, such as communities, public transport service levels, and management needs. In this paper, the three elements of density (building, population), function (land property, mixing degree, management needs), and form (three-dimensional height, body-shape coefficient, orientation) are used to assess travel behavior, car ownership, and building energy consumption to explore the relationship between the built environment and carbon emissions. Density factors include building density and population density, which are generally believed to be negatively correlated with carbon emissions [79,91,92]. In a city's central area, the density of buildings and population is large, which, combined with many employment opportunities and developed infrastructure and public transportation, reduces the scope of activities for residents to meet their work and living needs. Therefore, they will adopt a lower-carbon travel mode to reduce per capita energy consumption and emissions [93–95]. The functional elements include the nature of land use, land mixing degree, and managing demand. The functions of construction land are different. The corresponding building energy demand of buildings with different functions is different due to different use contents and intensity. The energy consumption of industrial land is generally higher than that of commercial and residential land [96,97]. In addition, the higher the mixing degree of buildings with different functions, the lower the carbon emissions [98]. Management needs usually involve the distance from the residence to the parking lot, the number of parking lots, and the level of parking services. Providing low-cost parking lots in the community will promote increased car ownership, increasing residents' dependence on cars and travel carbon emissions [99,100]. The morphological elements mainly include the three-dimensional urban structure and height, shape coefficient, and orientation of the building.

In recent years, urban buildings have continued to develop vertically. Although the compact distribution of buildings on the two-dimensional scale helps reduce $CO_2$ emissions, it may lead to more $CO_2$ emissions due to the dense and crowded patterns of urban structures in three-dimensional space, because high-density and high-rise urban three-dimensional structures reduce the air flow, sunlight, and ventilation capacity [101–103], leading to aggravation of the urban heat island phenomenon [104]. Furthermore, the power consumption of lighting and air conditioning will also increase significantly. Therefore, Lin et al. promote high-rise and low-density urban three-dimensional structures to reduce $CO_2$ emissions [63]. Furthermore, although high-rise buildings are conducive to increasing living density and reducing per capita carbon emissions, buildings of different heights change the morphological heterogeneity and vertical roughness of urban landscapes, lead-

ing to increased energy consumption and $CO_2$ emissions [105–108]. The building-shape coefficient refers to building surface area to volume. Cho (2002) found that the building-shape coefficient was positively correlated with carbon emissions when other conditions did not change [109]. Some researchers have also discussed the orientation of buildings. Different orientations mean different sunlight conditions, which directly affect the absorption or reflection of solar radiation on the building surface. Krüger et al. (2010) conducted a simulation study on the impact of building orientation on the carbon emissions of construction land in Israel's arid climate and found that the north–south orientation of the plots can produce shading effects between buildings, thereby reducing carbon emissions [110]. Therefore, the built environment's impact on carbon emissions is not a single factor but should be comprehensively considered based on multiple factors.

By considering the factors affecting the carbon emissions of the built environment, studies have shown that workplace density plays a more significant role in reducing $CO_2$ emissions than that of the home [111]. Therefore, reasonably increasing the building density of workplaces is one of the measures that contributes to low-carbon development. Secondly, based on the self-selection effect, residents could choose a built environment suitable for themselves according to their preferences [112,113], allowing residents who like to walk to choose walkable neighborhoods. Increasing walking frequency [114] allows residents with sustainable consumption concepts to live in communities close to public transportation and to reduce car ownership. In addition, it is necessary to pay attention to the reasonable layout and planning of parking lots, reduce unnecessary waste of land resources, and reduce the frequency of residents' private car use. Finally, the driving effect of policy factors on various functional facilities of the urban built environment should also attract attention to reduce the built environment's impact on cities' carbon emissions from multiple perspectives and accelerate the construction of a low-carbon city.

### 3.5.3. Transportation Networks

Examination of transportation networks mainly considers accessibility (destination accessibility, bus station accessibility), connectivity, road network density, and road design and hierarchy. Accessibility, connectivity, and road network density are related to whether a city has a developed road traffic network and a complete public transportation system. These factors affect transportation carbon emissions by changing the urban traffic environment (vehicle speed, congestion, air quality) and residents' travel behavior (travel range, car ownership, and use) [115–117]. The accessibility of the destination generally includes the distance from the residence to the central business district (CBD) and work accessibility. The closer distance to the CBD or the workplace, and the lower driving mileage reduce the number of family vehicles [118–121].

Similarly, the distance between households and public transportation stations will also affect car ownership. Potoglou et al. (2008) found that, if there is a public transportation station within walking distance of residences [122], household car ownership will decrease. One study showed that a 10% increase in public transport accessibility can reduce travel-related greenhouse gas emissions by 5.8% [123]. Furthermore, some scholars have proposed that the accessibility of green spaces and water spaces can also reduce $CO_2$ emissions related to residential energy consumption [124]. It is generally believed that connectivity and road network density negatively correlate with carbon emissions. Studies have shown that street connectivity positively correlates with low-carbon travel modes such as walking and cycling. Therefore, higher connectivity means fewer carbon emissions.

Furthermore, many studies on road network density have been carried out. Hong et al. (2014) found that if the intersection density in urban areas increases by 100%, the carbon emissions of transportation can decrease by 31.2–34.4% [125]. Wang et al. (2017) described the coupling relationship between urban spatial structure and traffic organization through road network density and the traffic coupling factor [126]; they found that a higher road network density can enhance the accessibility and connectivity of urban traffic and improve the efficiency of urban transportation energy consumption by increasing the speed

of vehicles and reducing traffic congestion. In addition, the design and structure of roads also affect carbon emissions. Studies have shown that designing streets suitable for walking or cycling for community residents will help them to adopt more "low-carbon" travel modes [127]. Agarwal et al. (2018) proposed a bicycle-only highway scheme in Patna, India. After implementing this infrastructure, the share of bicycle trips increased from 32 to 48%, effectively reducing transportation carbon emissions [128].

### 3.5.4. Development Patterns

Sprawl development and compact development are two key development modes in urbanization. Sprawl development reflects the one-time development of cities with high dispersion and low density. In contrast, compact development reflects the spatial density of cities (population, housing, employment) and the concentration of infrastructure [129], which indirectly affect the level of carbon emissions in cities and residents' daily lives. The general view is that guiding the transformation of cities from disorderly sprawl to high-density and compact development is conducive to carbon emission reduction [74,130]. Sprawl development is a kind of incremental development. The scale of a city continues to increase with the development of economic and travel tools. Such extensive expansion will cause harm to the environment and the development of the city. Further, the expansion will occupy a large amount of natural green space, reduce natural carbon sinks, and result in longer commuting distances and greater commuting flow, thereby increasing urban energy consumption [131,132]. At the same time, urban sprawl will lead to longer infrastructure service lines, resulting in massive demand for infrastructure investment. As a kind of stock development, compact development aims to slow the trend of urban sprawl and carry out intensive planning of urban form. It is generally believed that compact development is negatively related to carbon emissions [72,73,133] because a compact urban form helps reduce a city's erosion of the surrounding ecological environment and dependence on private cars. Improving the compactness of urban functions and forms, promoting the sharing of resources, services, and infrastructure, reducing land occupation by repeated construction, and reducing the energy and resource costs of urban operation are conducive to the sustainable development of a city. However, some scholars hold the opposite view, arguing that the benefits of compact development are not enough to make up for the adverse effects caused by it [108,134], such as road congestion, air pollution, low greening rate, and the heat island effect. Rong (2006) proposed a U-shaped relationship between compact development and energy consumption, with benefits increasing as density increases and decreasing as the urban heat island effect stimulates energy demand [135]. Although there is still controversy over the concept of compact development, this paper believes that the future direction is not only developing compact cities but how to develop more affordable and efficient compact cities

Due to the rigid needs of different types of urban development, sprawl and expansion are sometimes inevitable, but compact development is still the main direction. The two modes work together to form the complex urban form (single-center, multi-center), and proper planning of the development model is crucial for low-carbon development. Firstly, the development of compact cities is necessary. However, in the development process, attention should be paid to ensuring that the functions and forms of cities are compact, which is conducive to the complementarity between different functional units of urban space and reduces the commuting volume of urban residents [136]. Secondly, it is necessary to comprehensively consider the problems existing in single-center and multi-center development models [74,75,137] such as traffic congestion, poor greening, and high housing prices caused by single-center development [72,73]. Furthermore, multi-center development weakens the benefits created by public transportation, increasing infrastructure investment [138]. Based on these issues, we should explore a combination of the single-center development and multi-center development in the future. i.e., a balanced state to improve residents' living experience while reducing energy consumption. In addition, the impact of urban development models in cities at different development stages varies. In order to promote

low-carbon development, it is necessary to pay attention to adopting appropriate development models according to the constraints of cities at different development stages [139]. In a mega-city with a relatively high level of development, the development environment is the main restrictive factor (e.g., greenery coverage). It is necessary to focus on constructing various green spaces in compact development. Living consumption is the main limiting factor for coastal cities with a medium level of low-carbon development. Therefore, in a compact design, it is necessary to fully regulate residents' consumption and cultivate a low-carbon lifestyle. For inland cities with a low level of low-carbon development, there are constraints on economic development and social progress, so further optimization of industrial structure and improvement of energy utilization efficiency must be considered in urban development planning [140–142].

To summarize, the urban form includes main four aspects: land use, built environment, transportation network, and development pattern, which have direct or indirect effects on $CO_2$ emissions. However, the following problems also exist. Firstly, due to the complexity of influencing factors and the difference in the selection (e.g., land-use policy and development mode), a comprehensive systematic analysis framework and standardized index system have not been formed. Secondly, the current empirical research scales are concentrated in the urban dimension or the regional category, and there are few studies at the micro-scale (e.g., the community scale). Empirical research in different dimensions needs to be enriched. At the same time, although measures such as improving energy utilization efficiency and optimizing land-use structure have had positive effects on carbon emission reduction, they cannot fundamentally solve the problem of massive greenhouse gas emissions. Therefore, it is of great value to continuously reveal the potential impact factors of $CO_2$ emissions while comprehensively considering the urban spatial form, improving traffic efficiency, predicting the spatial distribution trend of $CO_2$, and examining other optimization aspects. The built environment is a city's backbone, and the transportation network represents its veins. Therefore, based on reasonable land use and the development model, we should first focus on building a comprehensive standardized index system. Next, carbon emission reduction strategies should be promoted while optimizing planning decisions to avoid weakening or offsetting benefits. Finally, all of the suggestions in this paper are considered to be of great significance to the realization of healthy and low-carbon development of cities.

## 4. Conclusions and Discussion

### 4.1. Conclusions of the Current Situation

This paper presents a visualization analysis of knowledge map of the correlation research between urban form and carbon emissions using the bibliometric method. It summarizes the characteristics of published papers, keywords, and co-citations. In addition, this paper examines the trends of popular research and provides an overview of current research. The conclusions of the paper include the following aspects:

(1) The research on the correlation between urban form and carbon emissions shows, overall, continuous growth in the total amount of literature, which can be divided into two development stages: the "development phase" and the "rapid development phase." In addition, carbon emissions are both a global and a regional issue. The process of urbanization occurred earlier in developed countries, and the United States and Europe paid earlier attention to the urban ecological environment. With the deepening of economic globalization, developing countries begin to pay increasing attention to the urban environment.

(2) Through the visual analysis of keywords, it was found that those with high popularity are carbon emission, urban form, sustainable, land use, and energy efficiency. In addition, the research can be divided into three stages of "budding-development-maturity" according to the degree of emergence.

(3) The impact of urban form on carbon emissions is a complex process of multiple factors. First, land-use structure, land-use efficiency, and land-management policies are the main land factors affecting carbon emissions. Second, the built environment affects

carbon emissions mainly through the combined action of three elements: density, function, and form. Furthermore, traffic accessibility, connectivity, and road network density can also affect transportation carbon emissions by changing the urban traffic environment and residents' travel behavior. Furthermore, urban development models have different impacts on carbon emissions of cities at different development stages, but compact urban development models are the main development direction.

*4.2. Discussion*

Based on the current research and practice on the correlation between urban form and carbon emissions, many studies have analyzed the correlation between their different factors., but lack a systematic research framework, an evaluation system, and micro-analysis. The urbanization process of most developing countries is still advancing rapidly. Determining how to optimize the urban form, promote energy conservation and emission reduction, and accelerate the development of low-carbon cities still faces severe challenges.

4.2.1. Build a Systematic Research Framework and Evaluation System

With the continuous increase in the urban population and the rapid development of the urban form, the methods to study the impact of urban form and carbon emissions have become complex and diverse. Some researchers have analyzed them through indicators such as accessibility and spot coverage. In contrast, others have used indicators such as production specialization and mobility, all of which have obtained credible results. However, the lack of systematic analysis, poor standardization of evaluation indicators, and the low availability of open data are related to the complexity of factors and the privacy of required data. Therefore, considering the requirement of low-carbon city construction in the future, it is necessary to build a systematic research framework to analyze the correlation between urban form and carbon emissions. Furthermore, it is also necessary to standardize the indicators and data to improve the research on urban carbon emissions, and the theory will be more comprehensive [143–145].

4.2.2. Microscopic Research

It can be found in the existing literature that the research scale of most works on the impact of urban form and carbon emissions is concentrated at the provincial level or the level of multiple urban agglomerations. However, few literature studies have examined the micro-dimension of cities, such as the street level or the community level. Although related research (zero-carbon community construction) has received attention in recent years, it is still necessary to explore the impact mechanism of urban form and carbon emissions at the micro-level. The joint development of macro-scale and micro-scale research reveals the impact mechanism more scientifically and effectively.

4.2.3. Method Optimization

With the development of RS and GIS, carbon satellites have gradually become a more applicable monitoring and accounting method. Although there are still deficiencies in the current technology, with the continuous development of the Global Carbon Assimilation System, assimilating satellite monitoring data and ground measurement data to reduce data errors can promote the rapid development of research on urban carbon emissions in the future. In addition, it is necessary to explore the combinations of new methods in the computation field, such as machine-learning algorithms, deep learning, and big data, with traditional analysis methods, to improve the efficiency and accuracy of analyses and accelerate the construction of sustainable urban forms that will meet various challenges in the future and promote the low-carbon development of cities.

**Author Contributions:** Conceptualization, C.S. and S.W.; methodology, Y.Z. and W.M.; software, Y.Z.; validation, Y.Z.; formal analysis, Y.Z.; investigation, C.S.; resources, C.S.; data curation, Y.Z.; writing—original draft preparation, Y.Z.; supervision, S.W. and R.W.; project administration, S.W.; funding acquisition, C.S. and R.W. All authors have read and agreed to the published version of the manuscript.

**Funding:** This work was supported by the National Natural Science Foundation of China (no. 41501184, no. 42001147), Technology Projects of Zhejiang Province (no. 2022C03168), Guangzhou Science and Technology Program (no. 202102020319) and Guangdong Province Natural Science Fund (no. 2022A1515011728).

**Institutional Review Board Statement:** Not applicable.

**Informed Consent Statement:** Not applicable.

**Data Availability Statement:** Data is contained within the article.

**Conflicts of Interest:** The authors declare no conflict of interest.

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
