# Peer review of "The Impacts of Urban Form on Carbon Emissions: A Comprehensive Review"

_land, doi:10.3390/land11091430_

Round 1
Reviewer 1 Report
The Authors justified the importance and necessity of conducting their research. However, the purpose of the presented work has not been clearly formulated. The data and methods do not raise any objections. The number of articles included in the research, nearly 2,500, is sufficient, also their time range, 2002-2021 and its length, are appropriate.
I believe that this article contains new information and valuable elements, but requires a decision as to its final content. Should it be a bibliometric article, a review of current research results, or both? Regardless of the outcome of this decision, the "Discussion" requires a thorough revision.
Detailed comments:
Fig 3. - some countries on the map are marked in white - please add information, what does white color mean - is there no data, or maybe the number of thematic publications in these countries was below the scale (<10)? Additionally, the names of some countries entered on the map are not their official names - please unify them (eg "America", "Britain" are colloquial names). Authors may also enter the accepted abbreviations of the full official names of selected countries and add an explanation below the figure.
Line 65 and line 73 - the numbering of two different subsections is identical, "2.1."
Line 91 - the marking of subsection 3.3.1 is incorrect, because the preceding section is numbered 3.1. Here the designation should be 3.1.1 and the next subsection 3.1.2. (line 98)
Lines 181-187: this is mostly a repetition of the wording from the introduction, I propose to delete it.
Discussion: I have no objections to lines 240-246, while the remainder of the text is not a discussion of the results of the authors' research, but an analysis of the results obtained by other authors. I believe that subsections 4.1. Methods of accounting for carbon dioxide emissions, 4.2.1. Land development, 4.2.2. The built environment, 4.2.3. Transport networks, 4.2.4 Patterns of development, are in fact not a discussion of the results presented by the authors in chapter 3. This part of the work could form the basis of a separate review article. Alternatively, these subchapters can be moved to chapter 3. "Results", to a new subchapter devoted to the analysis of current research results in selected articles. Therefore, in my opinion, the article must be rebuilt and chapter 4 "Discussion" needs to be rewritten.
Author Response
22-August-2022
Dear Editor,
We are very grateful for having a chance to improve our manuscript “The impacts of urban form on carbon emissions: A comprehensive review” (Land-1879334). We also appreciate the editors and reviewers who reviewed our research and paid so much patience to our manuscript; the detailed comments and suggestions are very significant and helpful for the authors to improve the research.
Based on the comments and suggestions, we have made careful modifications to the manuscript. The main corrections in the paper and the responds to the reviewers’ comments are appended below. The detailed information can also be seen in our revised manuscript. Revised portions are marked in red color in the revised paper.
Although the authors have carefully improved the paper in accordance with the comments and suggestions, there may still exist some problems and errors in our revised manuscript. We invite the editors and referees to propose more criticisms and suggestions. We also hope the new manuscript will meet Land’s standard with approval.
Best regards.
Yours sincerely,
The Authors.
Response to Reviewers
Land-1879334
The impacts of urban form on carbon emissions: A comprehensive review
- Response to Reviewer #1:
The Authors justified the importance and necessity of conducting their research. However, the purpose of the presented work has not been clearly formulated. The data and methods do not raise any objections. The number of articles included in the research, nearly 2,500, is sufficient, also their time range, 2002-2021 and its length, are appropriate. I believe that this article contains new information and valuable elements, but requires a decision as to its final content. Should it be a bibliometric article, a review of current research results, or both? Regardless of the outcome of this decision, the "Discussion" requires a thorough revision. Detailed comments:
(1) Fig 3. - some countries on the map are marked in white - please add information, what does white color mean - is there no data, or maybe the number of thematic publications in these countries was below the scale (<10)? Additionally, the names of some countries entered on the map are not their official names - please unify them (eg "America", "Britain" are colloquial names). Authors may also enter the accepted abbreviations of the full official names of selected countries and add an explanation below the figure.
Response: Thank you very much for your remind. We have carefully edited the figure 3 and made it more concise to improve its readability, like the meaning of the white color part in the map and names of some countries. The relevant content is modified as follows.
Figure 3. Spatial distribution of global publications.
(2) Line 65 and line 73 - the numbering of two different subsections is identical, "2.1." and Line 91 - the marking of subsection 3.3.1 is incorrect, because the preceding section is numbered 3.1. Here the designation should be 3.1.1 and the next subsection 3.1.2. (line 98).
Response: Thank you very much for your reminder. We have revised the serial number of the chapter and re-checked it carefully.
(3) Lines 181-187: this is mostly a repetition of the wording from the introduction, I propose to delete it.
Response: Thank you very much for your suggestion. We have deleted it.
(4) Discussion: I have no objections to lines 240-246, while the remainder of the text is not a discussion of the results of the authors' research, but an analysis of the results obtained by other authors. I believe that subsections 4.1. Methods of accounting for carbon dioxide emissions, 4.2.1. Land development, 4.2.2. The built environment, 4.2.3. Transport networks, 4.2.4 Patterns of development, are in fact not a discussion of the results presented by the authors in chapter 3. This part of the work could form the basis of a separate review article. Alternatively, these subchapters can be moved to chapter 3. "Results", to a new subchapter devoted to the analysis of current research results in selected articles. Therefore, in my opinion, the article must be rebuilt and chapter 4 "Discussion" needs to be rewritten.
Response: Thank you very much for your suggestion. We have moved Section “4.2” to Chapter 3 "Results" as a separate section “3.5. Impacts of urban form on carbon emissions”, and moved Section “4.1” to Chapter 3 "Results" as a separate section 3.4.1. In addition, the original section 5.2 of this paper is regarded as the “Discussion”, for the reason that the content of this part involves our introspection and outlook on the existing literature research. And we also make some change of the content of Conclusion. Hopefully this will make the overall structure of the article more reasonable. The relevant content is modified as follows.
(1) The research on the correlation between urban form and carbon emissions shows overall continuous growth in the total amount of literature, which can be di-vided into two development stages: the “development phase” and the “rapid development phase.” In addition, carbon emissions are both a global and a regional issue. The process of urbanization occurred earlier in developed countries, and the United States and Europe paid attention to the urban ecological environment earlier. With the deepening of economic globalization, developing countries begin to pay more and more attention to urban environment.
(2) Through the visual analysis of keywords, it is found that those with high popularity are “carbon emission,” “urban form,” “sustainable,” “land use,” and “energy efficiency.” At the same time, the research can be divided into three stages of “bud-ding-development-maturity” according to the degree of emergence.
(3) The impact of urban form on carbon emissions is a complex process of multiple factors. First of all, land use structure, land use efficiency and land management policies are the main land factors affecting carbon emissions. Secondly, the built environment affects carbon emissions mainly through the combined action of three elements: density, function, and form. Meanwhile, traffic accessibility, connectivity and road network density can also affect transportation carbon emissions by changing the urban traffic environment and residents' travel behavior. Furthermore, urban development models have different impacts on carbon emissions of cities at different development stages, but compact urban development models are the main development direction.
- Response to Reviewer #2:
(1) - Rows 45-51, the duality between urban form and urban development ought to be clearly/ better explained to the readers; as definition urban form (rows 45-47) is described as a process (Georgescu-Roegen, 1971, chapter 9 especially), with spatial and temporal borders (rows 456-7; the other explanations could to sustained this approach; the second line is to explain the urban form as a structure of (urban) characteristics (rows 444-446), therefore form(s) inside of the (same) unique process before described; this is the approach found, for example, at the rows 256-9).
-The issue of the urban form concept is not discussed; the initial concept of ” the spatial configuration of urban land use within a metropolitan area” (Chen et al., 2011, in References) is focused on the third direction, as the spatial configuration; în some parts of papers, this concept seems to be avoided (is missing): ” The keywords are divided into 12 clusters, and we mainly consider those associated with urban carbon emissions” (rows 135-6); also, the choice of this concept is not explained between/ from the similar urban concepts - in fact the (main) concept about all urban or settlement areas; as we know the urban area as settlement areas, has a large contribution to the carbon emissions; the city approach could be understood by the readers as avoiding analysing the towns and suburbs and rural areas – described by the three -level world classification for urban area.
Response: Thank you very much for your suggestion. We have carefully read the article you shared and agree with your definition of urban form. We think the duality of urban form and urban development can be explained from the relationship between the whole and the part. First of all, the both are a "Process". The urban development is the process of meeting the multi-level needs of the growing urban population, including quantitative expansion and qualitative improvement. Quantitative expansion is manifested in the increase of the number and scale of cities, that is, the improvement of the level of urbanization; the improvement in quality is manifested in the enhancement of urban functions and the improvement of the level of modernization. Urban form is also a process with the boundaries of space and time. Within the boundaries of time and space, urban form in a broad sense can be divided into two parts: tangible form and intangible form. The former mainly includes the built environment in the urban area, the characteristics of land use, and the differentiation pattern of various functions in the city. The latter refers to the spatial distribution of the city’s social, culture and other intangible elements. Secondly, we believe that the core of urban development lies in the process of development (such as land scale, population density, economic level), and the core of urban form lies in the change of form (such as spatial structure, social stratification), so the relationship between urban development and urban form is the duality which lies in its different core concerns.
In addition, we also try to understand urban form and urban development through the concept of “whole” and “part” in Georgescu-Roegen’s book. Urban development can be understood as a macroscopic whole, and the impact of urban form on carbon emissions in this paper focuses on urban form. The specific angle is a “part”, we agree with Chen’s point of view, but the urban form in this paper is not only the spatial configuration, but mainly includes the built environment, land use, transportation network and development mode, and the research scope is wider than Chen's. Our understanding is that the built environment and land use are the backbone of the city, the transportation network is the blood vessel of the city, and the development model is the brain of the city. In addition, it can be more effectively understood the impact of urban form on carbon emissions from the quantitative research on urban form with some indicators (such as accessibility, compactness, complexity). The quantitative research is beneficial to the development of the macro subject of "urban development".
Furthermore, our main research area in urbanized areas does not include suburban and rural areas, because urbanized areas is more representative, for the reason that the impact of carbon emissions in urbanized areas is much higher than that in rural areas. Thank you for your reminder, we have added explanations in the text. The relevant content is modified as follows.
Urban form can define as a process with temporal and spatial boundaries, including the spatial organization and arrangement of human activities, and at the same time it can affect the development and expansion of the city and the efficiency of resource al-location, land use, transportation and infrastructure, and promote city development better.
The keywords are divided into 12 clusters, and we mainly consider those associated with urban settlement areas carbon emissions, because the urban area as settlement areas, has a large contribution to the carbon emissions.
(2) - the assertion is not proved; also, the scale for Figure 6 is not inserted; also, for other figures.
Response: Thanks for your suggestion. We have thought carefully about the suggestions you put forward, and added two relevant documents to prove our point. In addition, according to the original data, we remade the chart and marked the scale. We hope this modification can reach your requirement. The relevant modification is as follows.
Figure 6. The top 25 keywords with the strongest citation burst words for the WOS. The data for this figure comes from Citespace.
(3) - Rows 234-6, “ Furthermore, while the estimation results of the measurement method are more accurate than the first two methods, this accuracy is dependent on local relevant data and the high cost of data collection.”; I suggest this assertion to be proved for the readers;
Response: Thank you very much for your suggestion. According to your suggestion, we searched relevant literature and proved this sentence. This article mainly discusses the shortcomings of the first two methods through the advantages and disadvantages of some data to prove it [1], and thanks again for the reminder here.
[1] Cai M., Shi Y., Ren C., Yoshida T., Yamagata Y., Ding C., Zhou N. The need for urban form data in spatial modeling of urban carbon emissions in China: A critical review. Journal of Cleaner Production, 2021, 139, 128792.
(4) - Rows 234-241, the switch between the described methods and the research methods used by the authors is not analysed for the readers.
Response: Thank you very much for your suggestion. After reading the literature, we found that some scholars believe that the accounting boundary of carbon emissions is mainly carried out from the production side, the consumption side and the "extraction emission" [1], and some scholars have proposed more specific accounting boundaries. That is the geographical boundary, the temporal boundary, the activity boundary and the lifecycle boundary [2]. These demarcation theories are the basis of carbon emission accounting methods, so we have summed up three accounting methods. We have supplemented the theory of carbon emission accounting in the article. At the same time, the accounting methods we propose are served to explore the impact of urban form on carbon emissions, because we focus on the latter, the previous explanation of carbon emissions accounting methods is insufficient. Thank you very much for your reminder to make the article more reasonable and complete.
[1] Kennedy S., Sgouridis S. Rigorous Classification and Carbon Accounting Principles for Low and Zero Carbon Cities. Energy Policy, 2011, 39(9), 5259-5268.
[2] Davis S. J., Peters G. P., Caldeira K. The Supply Chain of CO2 Emissions. Proceedings of the National Academy of Sciences, 2011, 108(45), 18554-18559.
(5) - Rows 528-9, the assertion ought to be discussed before Section 4; I suggest for example Dogaru et al. (2019).
Response: Thank you very much for the remind. We have carefully listened to your suggestions and partially adjusted the structure of the article according to the content we changed. Now we have adjusted it from the original 5 sections to 4 sections. We hope that such adjustment can be more reasonable.
(6) - I suggest the list with all papers used as a database be inserted into an Annex.
Response: Thank you very much for your suggestion. We have a large number of documents, and it will take a certain amount of time to sort them out and upload. If readers need it, we welcome readers contact us for it later.
- Response to Reviewer #3:
This manuscript (land-1879334) makes a very interesting literature review because: (1) the large number of related manuscripts that are considered and (2) the relevance of the topic to this distinguished journal: land use, urban form, and carbon emissions. The link between scientific research and practice is still missing and I therefore appreciate the efforts.
(1) The differences between this new review and some previous related reviews may need to be discussed.
Response: Thanks for your suggestion. After reading these review papers you shared with us, we found that the early review papers initially explored the correlation between urban form and energy consumption, but because of the limited number of empirical studies at that time, there was no systematic analysis of various factors affecting urban form carbon emissions. And the discussion also lacks a more systematic classification. Relatively new review papers only focus on one aspect of the impact of urban form on carbon emissions, such as the impact of urban form on household travel carbon emissions, few literatures comprehensively consider the comprehensive impact of urban form on carbon emissions, so some possibility may be ignored. In general, although previous papers have studied the relationship between urban form and carbon emissions from multiple perspectives, few papers have made a more comprehensive and systematic summary.
Our paper discusses the impact of urban form on carbon emissions, including transportation carbon emissions and building carbon emissions, etc., and organizes the current carbon emissions accounting methods and research methods to explore the correlation between the two (thanks again for your suggestions of adding three-dimensional urban structure), therefore, we hope that this review article is more comprehensive and systematic than previous articles. The relevant modification is as follows.
Although there are many studies on the correlation between urban form and carbon emissions, the previous review literature expounds the concept of urban form and the correlation between energy consumption, but there are few literatures discussing the influencing factors of urban form and carbon emissions [13], and some reviews only discuss the impact of urban form on transportation carbon emissions [14]. This paper not only considers the carbon emissions of transportation, but also considers the im-pact of urban morphological factors such as the built environment and three-dimensional urban structure on the carbon emissions of buildings, and also summarizes some commonly used research methods, so it is more systematic and comprehensive than previous reviews.
(2) Line 54-55: Why the impact of urban form on carbon emission has not yet been systematically and accurately quantified? Did the previous studies ignore this part?
Response: Thanks for your suggestion. First of all, there are some mistakes in our statement, which has been revised to "The quantitative research on the impact of urban form on carbon emissions needs to be further improved", for the reason that we have found through reading many literature that there are still some problems in the process of quantitative research on the impact of urban form on carbon emissions, though the accuracy of quantitative research has been continuously improved through some technical means and the improvement of data availability.
On the one hand, some quantitative studies only focus on a certain aspect of the impact of urban form on carbon emissions, lacking systematic [1-2].On the other hand, quantitative research indicators are diverse, such as density, scatter, leapfrogging, interspersion, accessibility etc., if different authors choose different indicators for quantitative research, the conclusions drawn may be much different or even diametrically opposite .For example, an article found that irregular patterns of urban structure would accelerate more carbon dioxide emissions due to the potential energy demand for transportation [3], but later concluded the opposite, and the explanation for this difference may be the previous the research does not consider the influence of 3D urban building index factors [4]. Therefore, we believe that there is still a lack of a systematic quantitative indicator research framework, which requires follow-up research to continuously improve quantitative research and build a more accurate and systematic research system.
[1] Zhang W., Huang B., Luo D. Effects of land use and transportation on carbon sources and carbon sinks: A case study in Shenzhen, China. Landscape and Urban Planning, 2014, 122, 175-185.
[2] Zuo S., Dai S., Ren Y. More fragmentized urban form more CO2 emissions? A comprehensive relationship from the combination analysis across different scales. Journal of Cleaner Production, 2020, 244, 118659.
[3] Ou J., Liu X., Li X., Chen Y. Quantifying the relationship between urban forms and carbon emissions using panel data analysis. Landsc. Ecol, 2013, 8 (10), 1889-1907.
[4] Xu X., Ou J., Liu P., Liu X., Zhang, H. Investigating the impacts of three-dimensional spatial structures on CO2 emissions at the urban scale. Sci. Total Environ. 2021, 762, 143096.
(3) I wonder why the authors started the bibliometric analysis from 2002?
Response: Thanks for your suggestion. When searching the literature, we found that the number of literatures before 2002 was too few, so we chose to start from 2002.
(4) Section 2.1: Some other similar terms, such as "urban configuration" and "spatial pattern", etc. could also be taken into consideration.
Response: Thanks for your suggestion. By searching and analyzing the literature, we found that on one hand, by comparing our content, we found that some categories in the text can include these two words such as built environment, land use, and development patterns have similar meanings to these two words, so we try to explain them in conjunction with our content. On the other hand, the few numbers of literatures can be searched by these two terms and cannot cover a lot of research content. I hope this modification can make the article more reasonable.
(5) In the Discussion Section, it is suggested that an important part of urban form, three-dimensional urban structure, should also be discussed (see below for example).
Response: We carefully considered your comments about the three-dimensional urban structure,and we add the impact of the three-dimensional urban structure in the subsection 3.5.2 “Built environment”, and It is discussed as part of the morphological elements of the urban form in Line 350-359. The relevant modification is as follows.
The morphological elements mainly include the three-dimensional urban structure and height, shape coefficient, and orientation of the building.
In recent years, urban buildings have continued to develop vertically. While the compact distribution of buildings on the two-dimensional scale helps reduce CO2 emissions, it may lead to more CO2 emissions due to the dense and crowded patterns of urban structures in three-dimensional space, because high-density and high-rise urban three-dimensional structures reduce the air flow, sunlight and ventilation ca-pacity [101-103], leading to aggravation of the urban heat island phenomenon [103-104], meanwhile, the power consumption of lighting and air conditioning will al-so increase significantly [80]. Therefore, Lin et al. promote high-rise and low-density urban three-dimensional structures to reduce CO2 emissions.
(6) It would be better if this manuscript could further summarize the methods (e.g., regression, correlation) used for analyzing the relationships between urban form and carbon emissions.
Response: We are very glad that you can give us this one suggestion. We have carefully listened to your suggestions. By reading a lot of literature, we divide the existing research methods into spatial methods and non-spatial methods. Spatial methods mainly include geographic weighting models, spatial autocorrelation models, and GIS spatial analysis models, etc. Non-spatial methods mainly include linear or nonlinear correlation models, panel data models and index decomposition simulation methods (such as Kaya identity model and STIRPAT model). And we also introduce the more cutting-edge methods such as random forest regression models. It made our article structure more complete. Thank you again for your professional suggestion. The relevant modification is as follows.
The current research methods of urban form and carbon emissions can be mainly divided into spatial methods and non-spatial methods. Spatial methods mainly use geographic information or spatial distribution data to explore the spatial differentiation and spillover effect or the impact mechanism of carbon emissions, including the geographic weighting models, the spatial autocorrelation models and the GIS spatial analysis etc. [55-57]. For example, Wang et al. used spatial autocorrelation model to explore the spatial spillover effect and driving force of China's urban carbon emission [58]. Meanwhile, the non-spatial types of data (eg, the economic status, the population density, the energy consumption) is frequently used to identify the impact of different factors of urban form on carbon emissions in the non-spatial methods, which contains the linear or the nonlinear correlation models, the panel data models and the index decomposition methods (eg, Kaya identity model and STIRPAT model) [53-54]. [59-60]. For example, Yang et al. found that China's urban form and traffic factors have a significant impact on per capita carbon dioxide emissions through a panel data model [61]. Van et al. used a multiple linear regression model to study the impact of the built environment on residents' travel carbon emissions, and found that the average travel time was negatively correlated with rail transit density [62].
With the improvement of data acquisition capabilities and the development of technology, new methods are constantly emerging. To avoid the negative consequences of multicollinearity between explanatory variables and improve the accuracy of the results, the random forest regression mode based on machine learning is widely ap-plied in research. Lin et al. found through a random forest regression model that the degree of spatial crowding plays a dominant role in the variation of CO2 emissions [63]. In short, there are many types of research methods on the correlation between urban form and carbon emissions, which need to be appropriately selected according to different needs to ensure the reasonable of the results.
*********************************************************************
References
- Masson-Delmotte, V.; Zhai, P.; Pirani, A.; Connors, S. L.; Pean, C.; Berger, S. ; Zhou, B. Climate change 2021: the physical science
basis. Contribution of working group I to the sixth assessment report of the intergovernmental panel on climate change, 2.
- Ritchie, H.; Roser, M. Our world in data. 2018
- Madlener, R.; Sunak, Y. Impacts of urbanization on urban structures and energy demand: What can we learn for urban energy planning and urbanization management?. Sustainable Cities and Society. 2011, 1(1), 45–53.
- Georgescu-Roegen, N. The Entropy Law and the Economic Process,1971.
- Chen, Y.; Li, X.; Zheng, Y.,; Guan, Y.; Liu, X. Estimating the relationship between urban forms and energy consumption: A case study in the Pearl River Delta, 2005–2008. Landscape and urban planning. 2011, 201(1), 33–42.
- He, S.; Yu, S.; Li, G.; Zhang, J. Exploring the influence of urban form on land-use efficiency from a spatiotemporal heterogeneity perspective: Evidence from 336 Chinese cities. Land Use Policy, 2020, 95, 104576.
- Lee, J.; Kurisu, K.; An, K.; Hanaki, K. Development of the compact city index and its application to Japanese cities. Urban Studies, 2008, 52(6), 1054–1070.
- Guo, C.; Schwarz, N.; Buchmann, C. M. Exploring the added value of population distribution indicators for studies of European urban form. Applied spatial analysis and policy, 2018, 11(3), 439–463.
- Kaza, N. Urban form and transportation energy consumption. Energy Policy, 2020, 136, 111049.
- Zhang, W.; Huang, B.; Luo, D. Effects of land use and transportation on carbon sources and carbon sinks: A case study in Shenzhen, China. Landscape and Urban Planning, 2014, 122, 175–185.
- Wang, S.; Li, G.; Fang, C. Urbanization, economic growth, energy consumption, and CO2 emissions: Empirical evidence from countries with different income levels. Renewable and Sustainable Energy Reviews, 2018, 81, 2144–2159.
- Wang, S.; Wang, J.; Fang, C.; Li, S. Estimating the impacts of urban form on CO2 emission efficiency in the Pearl River Delta, China. Cities, 2019, 85, 117–129.
- Anderson,; W, Kanaroglou, P,; Miller,E. Urban form, energy and the environment: a review of issues, evidence and policy. Urban Studies, 1996,33(1): 7-35
- Tian, X.; An, C.; Chen,Z. Assessing the impact of urban form on the greenhouse gas emissions from household vehicles: a review. Journal of Environmental Informatics Letters, 2020, 3(2).
- Boyack, K. W.; Wylie, B. N.; Davidson, G. S. Information visualization, human-computer interaction, and cognitive psychology: domain visualizations. Springer, Berlin, Heidelberg, 2002,In Visual Interfaces to Digital Libraries (pp. 145-158).
- Chen, C.; Song, M. Visualizing a field of research: A methodology of systematic scientometric reviews. PloS one, 2009, 14(10), e0223994.
- Van Eck, N. J.; Waltman, L.; Dekker, R.; Van Den Berg, J. A comparison of two techniques for bibliometric mapping: Multidimensional scaling and VOS. Journal of the American Society for Information Science and Technology, 2010, 61(12), 2405-2416.
- Yang, W.; Zhang, J.; Ma, R. The prediction of infectious diseases: A bibliometric analysis. International Journal of Environmental Research and Public Health, 2020, 17(17),
- Hu, Y.; Yu, Z.; Cheng, X.; Luo, Y.; Wen, C. A bibliometric analysis and visualization of medical data mining research. Medicine , 2020, 99(22).
- Xie, H.;Wen, Y.; Choi, Y.; Zhang, X. Global Trends on Food Security Research: A Bibliometric Analysis. Land, 2008, 10(2), 119.
- Sharifi, A. Urban resilience assessment: Mapping knowledge structure and trends. Sustainability, 2020, 12(15), 5918.
- Sharifi, A.; Allam, Z.; Feizizadeh, B.; Ghamari, H. Three decades of research on smart cities: Mapping knowledge structure and trends. Sustainability, 2021, 13(13), 7140.
- Wei, F.; Grubesic, T. H.; Bishop, B.W. Exploring the GIS knowledge domain using CiteSpace. The Professional Geographer, 2015, 67(3), 374-384.
- Lebel, L.; Garden, P.; Banaticla, M. R. N.; Lasco, R. D.; Contreras, A.; Mitra, A. P.; Sari, A. Integrating carbon management into the development strategies of urbanizing regions in Asia - Implications of urban function, form, and role. Journal of Industrial Ecology, 2008, 11(2), 61-81.
- Hankey, S.; Marshall, J. D. Impacts of urban form on future US passenger-vehicle greenhouse gas emissions. Energy Policy, 2010, 38(9), 4880-4887.
- Croci, E.; Melandri, S.; Molteni, T. Determinants of cities’ GHG emissions: a comparison of seven global cities. International Journal of Climate Change Strategies and Management, 2011, 3(3), 275-301.
- Liu, X.; Sweeney, J. Modelling the impact of urban form on household energy demand and related CO2 emissions in the Greater Dublin Region. Energy Policy, 2012, 46, 359-369.
- Bereitschaft, B.; Debbage, K. Urban form, air pollution, and CO2 emissions in large US metropolitan areas. The Professional Geographer, 2013, 65(4), 612-635.
- Gurney, K. R. Recent research quantifying anthropogenic CO2 emissions at the street scale within the urban domain. Carbon Management, 2014, 5(3), 309-320.
- Baiocchi, G.; Creutzig, F.; Minx, J.; Pichler, P. P. A spatial typology of human settlements and their CO2 emissions in England. Global Environmental Change, 2015, 34, 13-21.
- Marchi, M.; Pulselli, R. M.; Marchettini, N.; Pulselli, F. M.; Bastianoni, S. Carbon dioxide sequestration model of a vertical greenery system. Ecological Modelling, 2015, 306, 46-56.
- Ma, M.; Rozema, J.; Gianoli, A.; Zhang, W. The Impacts of City Size and Density on CO2 Emissions: Evidence from the Yangtze River Delta Urban Agglomeration. Applied Spatial Analysis and Policy, 2021, 1-27.
- Fernandes, M.; Skjemstad, J.; Johnson, B.; Wells, J.; Brooks, P.Characterization of carbonaceous combustion residues. I. Morphological, elemental and spectroscopic features. CHEMOSPHERE,2003, 51(8), 785-795.
- Arnfield, A. Two decades of urban climate research: A review of turbulence, exchanges of energy and water, and the urban heat island. International Jounal of Climatology,2003,23(1),1-26.
- Dhakal, S. Urban energy use and carbon emissions from cities in China and policy implications. Energy policy, 2009, 37(11), 4208-4219.
- Davis, S.; Peters, G.; Caldeira K. The Supply Chain of CO2 Emissions. Proceedings of the National Academy of Sciences, 2011, 108(45),18554-18559.
- Kennedy, S.; Sgouridis S. Rigorous Classification and Carbon Accounting Principles for Low and Zero Carbon Cities. Energy Policy,2011, 39(9), 5259-5268.
- IPCC, C. C. The physical science basis. Contribution of working group I to the fourth assessment report of the Intergovernmental Panel on Climate Change. Cambridge University Press, Cambridge, United Kingdom and New York, NY, USA, 996(2007), 113-119.
- Dodman, D. Blaming cities for climate change? An analysis of urban greenhouse gas emissions inventories. Environment and urbanization, 2009, 21(1), 185-201.
- Kennedy, C.; Steinberger, J.; Gasson, B.; Hansen, Y.; Hillman, T.; Havránek, M.; Mendez, G. V. Methodology for inventorying greenhouse gas emissions from global cities. Energy policy, 2010, 38(9), 4828-4837.
- Kennedy, C. A.; Ramaswami, A.; Carney, S.; Dhakal, S. Greenhouse gas emission baselines for global cities and metropolitan regions. Cities and climate change: Responding to an urgent agenda 2011, 15-54.
- Shan, Y.; Liu, J.; Liu, Z.; Xu, X.; Shao, S.; Wang, P.; Guan, D. New provincial CO2 emission inventories in China based on apparent energy consumption data and updated emission factors. Journal Abbreviation 2016, 184, 742-750.
- Cai, M.; Shi, Y.; Ren, C. Developing a high-resolution emission inventory tool for low-carbon city management using hybrid method–A pilot test in high-density Hong Kong. Energy and Buildings, 2020, 226, 110376.
- Chuai, X.; Feng, J. High resolution carbon emissions simulation and spatial heterogeneity analysis based on big data in Nanjing City, China. Science of The Total Environment, 2019, 686, 828-837.
- Elvidge, C. D.; Baugh, K. E.; Zhizhin, M.; Hsu, F. C. Why VIIRS data are superior to DMSP for mapping nighttime lights. Proceedings of the Asia-Pacific Advanced Network, 2013, 35(0), 62.
- Small, C.; Pozzi, F.; Elvidge, C. D. Spatial analysis of global urban extent from DMSP-OLS night lights. Remote Sensing of Environment, 2005, 96(3-4), 277-291.
- Lu, H.; Liu, G.; Miao, C.; Zhang, C.; Cui, Y.; Zhao, J. Spatial pattern of residential carbon dioxide emissions in a rapidly urbanizing Chinese city and its mismatch effect. Sustainability, 2018, 10(3), 827.
- Doll, C. N.; Muller, J. P.; Elvidge, C. D. Night-time imagery as a tool for global mapping of socioeconomic parameters and greenhouse gas emissions. Ambio, 2000, 157-162.
- Elvidge, C. D.; Keith, D. M.; Tuttle, B. T.; Baugh, K. E. Spectral identification of lighting type and character. Sensors, 2010, 10(4), 3961-3988.
- Cai, B.; Liang, S.; Zhou, J.; Wang, J.; Cao, L.; Qu, S.; Yang, Z. China high resolution emission database (CHRED) with point emission sources, gridded emission data, and supplementary socioeconomic data. Resources, Conservation and Recycling, 2018, 129, 232-239.
- Yamagata, Y.; Murakami, D.; Yoshida, T. Urban carbon mapping with spatial BigData. Energy Procedia, 2017, 142, 2461-2466.
- Boyd, J.; Budinov, D.; Robinson, I.; Jack, J. The measurement of carbon dioxide levels in a city canyon. In Remote Sensing Technologies and Applications in Urban Environments, 2016, (Vol. 10008, pp. 78-87). SPIE.
- Zhang, G.; Ge, R.; Lin, T.; Ye, H.; Li, X.; Huang, N. Spatial apportionment of urban greenhouse gas emission inventory and its implications for urban planning: A case study of Xiamen, China. Ecological Indicators, 2018, 85, 644-656.
- Cai, M.; Shi, Y.; Ren, C.; Yoshida, T.; Yamagata, Y.; Ding, C.; Zhou, N.The need for urban form data in spatial modeling of urban carbon emissions in China: A critical review. Journal of Cleaner Production,2021,139,128792
- Wang, S.; Shi, C.; Fang, C.; Feng, K. Examining the spatial variations of determinants of energy-related CO2 emissions in China at the city level using Geographically Weighted Regression Model. Applied Energy, 2019, 235: 95-105.
- Wang, Y.; Zhao, M.; Chen, W. Spatial effect of factors affecting household CO2 emissions at the provincial level in China: a geographically weighted regression model. Carbon Management, 2018, 9(2), 187-200.
- Hou, Q.; Zhang, X.; Li, B.; Zhang, X.; Wang, W. Identification of low-carbon travel block based on GIS hotspot analysis using spatial distribution learning algorithm. Neural Computing & Applications,2019, 31(9), 4703-4713.
- Wang, S.; Huang, Y.; Zhou, Y. Spatial spillover effect and driving forces of carbon emission intensity at the city level in China. Journal of Geographical Sciences, 2019, 29 (2), 231-252.
- Wu, Y.; Shen, J.; Zhang, X.; Skitmore, M.; Lu, W. The impact of urbanization on carbon emissions in developing countries: a Chinese study based on the U-Kaya method. Journal of Cleaner Production,2017, 163, S284-S298.
- Wang, P.; Wu, W.; Zhu, B.; Wei, Y. Examining the impact factors of energy-related CO2 emissions using the STIRPAT model in Guangdong Province, China. Applied Energy, 2013, 106,65-71.
- Yang, W.; Tao, L.; Cao, X. Examining the impacts of socio-economic factors, urban form and transportation development on CO2 emissions from transportation in China: A panel data analysis of China's provinces, Habitat International,2015,49, 212-220.
- Van D.; Schwanen T. Re-evaluating the impact of urban form on travel patterns in Europe and North-America. Transport Policy, 2006, 13(3): 229-239.
- Lin, J.; Lu, S.; He, X.; Wang, F. Analyzing the impact of three-dimensional building structure on CO2 emissions based on random forest regression. Energy,2021,236.
- Ewing, R.; Cervero, R. Travel and the built environment: A meta-analysis. Journal of the American planning association, 2010, 76(3), 265-294.
- Fong, W. K.; Matsumoto, H.; Ho, C. S.; Lun, Y. F. Energy consumption and carbon dioxide emission considerations in the urban planning process in Malaysia. Planning Malaysia, 2008, 6.
- Khan, F.; Pinter, L. Scaling indicator and planning plane: An indicator and a visual tool for exploring the relationship between urban form, energy efficiency and carbon emissions. Ecological Indicators, 2016, 67, 183-192.
- Pan, Y.; Birdsey, R. A.; Fang, J.; Houghton, R.; Kauppi, P. E.; Kurz, W. A.; Hayes, D. A large and persistent carbon sink in the world’s forests. science, 2011, 333(6045), 988-993.
- Seto, K. C.; Güneralp, B.; Hutyra, L. R. Global forecasts of urban expansion to 2030 and direct impacts on biodiversity and carbon pools. Proceedings of the National Academy of Sciences, 2012, 109(40), 16083-16088.
- Bateman, I. J.; Harwood, A. R.; Mace, G. M.; Watson, R. T.; Abson, D. J.; Andrews, B.; Termansen, M. Bringing ecosystem services into economic decision-making: land use in the United Kingdom. science, 2013, 341(6141), 45-50.
- Xu, Q.; Yang, R.; Dong, Y. X.; Liu, Y. X.; Qiu, L. R. The influence of rapid urbanization and land use changes on terrestrial carbon sources/sinks in Guangzhou, China. Ecological Indicators, 2016, 70, 304-316.
- Liu, J.; Kuang, W.; Zhang, Z.; Xu, X.; Qin, Y.; Ning, J.; Chi, W. Spatiotemporal characteristics, patterns, and causes of land-use changes in China since the late 1980s. Journal of Geographical sciences, 2014, 24(2), 195-210.
- Ou, J.; Liu, X.; Li, X.; Chen, Y. Quantifying the relationship between urban forms and carbon emissions using panel data analysis. Landscape ecology, 2013, 28(10), 1889-1907.
- Liu, X.; Wang, M.; Qiang, W.; Wu, K.; Wang, X. Urban form, shrinking cities, and residential carbon emissions: Evidence from Chinese city-regions. Applied Energy, 2020, 261, 114409.
- Shi, K.; Xu, T.; Li, Y.; Chen, Z.; Gong, W.; Wu, J.; Yu, B. Effects of urban forms on CO2 emissions in China from a multi-perspective analysis. Journal of Environmental Management, 2020, 262, 110300.
- Zhao, P.; Diao, J.; Li, S. The influence of urban structure on individual transport energy consumption in China’s growing cities. Habitat International, 2017, 66, 95-105.
- Muñiz, I.; Dominguez, A. The impact of urban form and spatial structure on per capita carbon footprint in US larger metropolitan areas. Sustainability, 2020, 12(1), 389.
- Lu, J.; Guldmann, J. M. Landscape ecology, land-use structure, and population density: Case study of the Columbus Metropolitan Area. Landscape and Urban Planning, 2012, 105(1-2), 74-85.
- Chuai, X.; Huang, X.; Wang, W.; Zhao, R.; Zhang, M.; Wu, C. Land use, total carbon emissions change and low carbon land management in Coastal Jiangsu, China. Journal of Cleaner Production, 2015, 103, 77-86.
- Glaeser, E. L.; Kahn, M. E. The greenness of cities: Carbon dioxide emissions and urban development. Journal of urban economics, 2010, 67(3), 404-418.
- Huang, Y.; Xia, B.; Yang, L. Relationship study on land use spatial distribution structure and energy-related carbon emission intensity in different land use types of Guangdong, China, 1996–2008. The Scientific World Journal, 2013, 2013.
- Wu, C.; Li, G.; Yue, W.; Lu, R.; Lu, Z.; You, H. Effects of endogenous factors on regional land-use carbon emissions based on the grossman decomposition model: a case study of Zhejiang province, China. Environmental management, 2015, 55(2), 467-478.
- Gratani, L.; Varone, L.; Bonito, A. Carbon sequestration of four urban parks in Rome. Urban Forestry & Urban Greening, 2016, 19, 184-193.
- Jo, H. K.; Kim, J. Y.; Park, H. M. Carbon reduction and planning strategies for urban parks in Seoul. Urban Forestry & Urban Greening, 2019, 41, 48-54.
- Wang, Y.; Chang, Q.; Li, X. Promoting sustainable carbon sequestration of plants in urban greenspace by planting design: A case study in parks of Beijing. Urban Forestry & Urban Greening, 2021, 64, 127291.
- Lv, F.; Qi, X. C. The Urban Planning Strategies Research of Decreasing Traffic Carbon Emissions. Trans Tech Publications Ltd 2013, In Advanced Materials Research (Vol. 671, pp. 2402-2405).
- Zhang, R.; Matsushima, K.; Kobayashi, K. Can land use planning help mitigate transport-related carbon emissions? A case of Changzhou. Land Use Policy, 2018, 74, 32-40.
- Grimm, N. B.; Faeth, S. H.; Golubiewski, N. E.; Redman, C. L.; Wu, J.; Bai, X.; Briggs, J. M. Global change and the ecology of cities. science, 2008, 319(5864), 756-760.
- Hutyra, L. R.; Yoon, B.; Hepinstall-Cymerman, J.; Alberti, M. Carbon consequences of land cover change and expansion of urban lands: A case study in the Seattle metropolitan region. Landscape and urban planning, 2011, 103(1), 83-93.
- Pickett, S. T.; Cadenasso, M. L.; Grove, J. M.; Boone, C. G.; Groffman, P. M.; Irwin, E.; Warren, P. Urban ecological systems: Scientific foundations and a decade of progress. Journal of environmental management, 2011, 92(3), 331-362.
- Cervero, R.; Kockelman, K. Travel demand and the 3Ds: Density, diversity, and design. Transportation research part D: Transport and environment, 1997, 2(3, 199-219.
- Deng, J. Y.; Wong, N. H.; Zheng, X. The study of the effects of building arrangement on microclimate and energy demand of CBD in Nanjing, China. Procedia engineering, 2016, 169, 44-54.
- Fang, C.; Wang, S.; Li, G. Changing urban forms and carbon dioxide emissions in China: A case study of 30 provincial capital cities. Applied energy, 2015, 158, 519-531.
- Cervero, R.; Radisch, C. Travel choices in pedestrian versus automobile oriented neighborhoods. Transport policy, 1996, 3(3), 127-141.
- McCormack, E.; Rutherford, G. S.; Wilkinson, M. G. Travel impacts of mixed land use neighborhoods in Seattle, Washington. Transportation Research Record, 2001, 1780(1), 25-32.
- Kuzmyak, J. R.; Baber, C.; Savory, D. Use of walk opportunities index to quantify local accessibility. Transportation Research Record 2006, 1977(1), 145-153.
- Song, Y. B. Influence of new town development on the urban heat island-The case of the Bundang area. Journal of Environmental Sciences (China), 2005, 17(4), 641-645.
- Weng, Q.; Lu, D.; Schubring, J. Estimation of land surface temperature–vegetation abundance relationship for urban heat island studies. Remote sensing of Environment, 2004, 89(4), 467-483.
- Masnavi, M. R. The new millennium and the new urban paradigm: the compact city in practice. Achieving sustainable urban form, 2000, 64-73.
- Guo, Z. Does residential parking supply affect household car ownership? The case of New York City. Journal of Transport Geography, 2013, 26, 18-28.
- Van, E.; Looman, R.; Bruin-Hordijk G. The effects of urban and building design parameters on solar access to the urban canyon and the po-tential for direct passive solar heating strategies. Energy Build,2012;47:189e200.
- Shi, Y.; Ren, C.; Zheng, Y.; Ng, E. Mapping the urban microclimatic spatial distribution in a sub-tropical high-density urban environment. Architect Sci Rev, 2016;59(5):370e84.
- Dawodu, A.; Cheshmehzangi, A. Impact of floor area ratio (FAR) on energy consumption at meso scale in China: case study of ningbo. Energy Procedia, 2017;105:3449e55.
- Perini, K.; Magliocco, A. Effects of vegetation, urban density, building height, and atmospheric conditions on local temperatures and thermal comfort. Urban For Urban Green, 2014;13(3):495e506.
- Coseo, P.; Larsen, L. How factors of land use/land cover, building configuration, and adjacent heat sources and sinks explain Urban Heat Islands in Chicago. Landsc Urban Plann, 2014;125:117e29.
- Tyrinopoulos, Y.; Antoniou, C. Factors affecting modal choice in urban mobility. European Transport Research Review, 2013, 5(1), 27-39.
- Yan, H.; Shen, Q.; Fan, L. C.; Wang, Y.; Zhang, L. Greenhouse gas emissions in building construction: A case study of One Peking in Hong Kong. Building and Environment, 2010, 45(4), 949-955.
- Godoy-Shimizu, D.; Steadman, P.; Hamilton, I.; Donn, M.; Evans, S.; Moreno, G.; Shayesteh, H. Energy use and height in office buildings. Building Research & Information, 2018, 46(8), 845-863.
- Borck, R. Will skyscrapers save the planet? Building height limits and urban greenhouse gas emissions. Regional Science and Urban Economics, 2016, 58, 13-25.
- Cho, J. S. Design methodology for tall office buildings: Design measurement and integration with regional character. Illinois Institute of Technology. 2002.
- Krüger, E.; Pearlmutter, D.; Rasia, F. Evaluating the impact of canyon geometry and orientation on cooling loads in a high-mass building in a hot dry environment. Applied energy, 2010, 87(6), 2068-2078.
- Ding, C.; Wang, Y.; Xie, B.; Liu, C. Understanding the role of built environment in reducing vehicle miles traveled accounting for spatial heterogeneity. Sustainability, 2014, 6(2), 589-601.
- Handy, S. Critical assessment of the literature on the relationships among transportation, land use, and physical activity. Transportation Research Board and the Institute of Medicine Committee on Physical Activity, Health, Transportation, and Land Use. Resource paper for TRB Special Report, 2005, 282(1), 1-81.
- Cao, X.; Mokhtarian, P. L.; Handy, S. L. Examining the impacts of residential self-selection on travel behaviour: a focus on empirical findings. Transport reviews, 2009, 29(3), 359-395.
- Handy, S. L.; Clifton, K. J. Local shopping as a strategy for reducing automobile travel. Transportation, 2001, 28(4), 317-346.
- Berdica, K.; Andjic, Z.; Nicholson, A. J. Simulating Road Traffic Interruptions–Does it Matter What Model We Use?. In The Network Reliability of Transport. Emerald Group Publishing Limited, 2003.
- Lu, Q. C.; Peng, Z. R. Vulnerability analysis of transportation network under scenarios of sea level rise. Transportation research record, 2011, 2263(1), 174-181.
- Taylor, M. A. Critical transport infrastructure in Urban areas: impacts of traffic incidents assessed using accessibility-based network vulnerability analysis. Growth and Change, 2008, 39(4), 593-616.
- Krizek, K. J. Residential relocation and changes in urban travel: Does neighborhood-scale urban form matter?. Journal of the American Planning Association, 2003, 69(3), 265-281.
- Miller, E. J.; Ibrahim, A. Urban form and vehicular travel: some empirical findings. Transportation Research Record, 1998, 1617(1), 18-27.
- Cervero, R.; Arrington, G. B. Vehicle trip reduction impacts of transit-oriented housing. Journal of Public Transportation, 2008, 11(3), 1.
- Van Acker, V.; Witlox, F. Car ownership as a mediating variable in car travel behaviour research using a structural equation modelling approach to identify its dual relationship. Journal of Transport Geography, 2010, 18(1), 65-74.
- Potoglou, D.; Kanaroglou, P. S. Modelling car ownership in urban areas: a case study of Hamilton, Canada. Journal of Transport Geography, 2008, 16(1), 42-54.
- Zahabi, S. A. H.; Miranda-Moreno, L.; Patterson, Z.; Barla, P.; Harding, C. Transportation greenhouse gas emissions and its relationship with urban form, transit accessibility and emerging green technologies: a Montreal case study. Procedia-Social and Behavioral Sciences, 2012, 54, 966-978.
- Ye, H.; He, X.; Song, Y.; Li, X.; Zhang, G.; Lin, T.; Xiao, L. A sustainable urban form: The challenges of compactness from the viewpoint of energy consumption and carbon emission. Energy and Buildings, 2015, 93, 90-98.
- Hong, J.; Goodchild, A. Land use policies and transport emissions: Modeling the impact of trip speed, vehicle characteristics and residential location. Transportation Research Part D: Transport and Environment, 2014, 26, 47-51.
- Wang, S.; Liu, X.; Zhou, C.; Hu, J.; Ou, J. Examining the impacts of socioeconomic factors, urban form, and transportation networks on CO2 emissions in China’s megacities. Applied energy, 2017, 185, 189-200.
- Holz-Rau, C.; Scheiner, J.; Sicks, K. Travel distances in daily travel and long-distance travel: what role is played by urban form?. Environment and Planning A, 2014, 46(2), 488-507.
- Agarwal, A.; Ziemke, D.; Nagel, K. Bicycle superhighway: An environmentally sustainable policy for urban transport. Transporta- tion Research Part A: Policy and Practice, 2020, 137, 519-540.
- Makido, Y.; Dhakal, S.; Yamagata, Y. Relationship between urban form and CO2 emissions: Evidence from fifty Japanese cities. Urban Climate, 2012, 2, 55-67.
- Ou, J.; Liu, X.; Wang, S.; Xie, R.; Li, X. Investigating the differentiated impacts of socioeconomic factors and urban forms on CO2 emissions: Empirical evidence from Chinese cities of different developmental levels. Journal of Cleaner Production, 2019, 226, 601-614.
- Zhang, Y.; Wu, Q.; Fath, B. D. Review of spatial analysis of urban carbon metabolism. Ecological Modelling, 2018, 371, 18-24.
- Cirilli, A.; Veneri, P. Spatial structure and carbon dioxide (CO2) emissions due to commuting: An analysis of Italian urban areas. Regional Studies, 2014, 48(12), 1993-2005.
- Lee, S.; Lee, B. The influence of urban form on GHG emissions in the US household sector. Energy Policy, 2014, 68, 534-LIU
- Liang, Z.; Wu, S.; Wang, Y.; Wei, F.; Huang, J.; Shen, J.; Li, S. The relationship between urban form and heat island intensity along the urban development gradients. Science of the Total Environment, 2020, 708, 135011.
- Ewing, R.; Rong, F. The impact of urban form on US residential energy use. Housing policy debate, 2008, 19(1), 1-30.
- Luo, Y. L.; Zhang, C. X. Research on the Low-Carbon Land Use Pattern. Trans Tech Publications Ltd, 2012, In Advanced Materials Research (Vol. 598, pp. 241-246).
- Wang, S.; Zeng, J.; Huang, Y.; Shi, C.; Zhan, P. The effects of urbanization on CO2 emissions in the Pearl River Delta: a comprehensive assessment and panel data analysis. Applied Energy, 2018, 228, 1693-1706.
- Wang, Y.; Hayashi, Y.; Chen, J.; Li, Q. Changing urban form and transport CO2 emissions: an empirical analysis of Beijing, China. Sustainability, 2014, 6(7), 4558-4579.
- Su, M.; Liang, C.; Chen, B.; Chen, S.; Yang, Z. Low-carbon development patterns: observations of typical Chinese cities. Energies, 2012, 5(2), 291-304.

Reviewer 2 Report
I attach the file.

Author Response

(The authors gave the same response as above.)

Reviewer 3 Report
This manuscript (land-1879334) makes a very interesting literature review because: (1) the large number of related manuscripts that are considered and (2) the relevance of the topic to this distinguished journal: land use, urban form, and carbon emissions. The link between scientific research and practice is still missing and I therefore appreciate the efforts. I just have some comments and suggestions presented as follows:
(1) The differences between this new review and some previous related reviews may need to be discussed.
Assessing the Impact of Urban Form on the Greenhouse Gas Emissions from Household Vehicles: A Review. Journal of Environmental Informatics Letters, 2020, 3(2)
Urban form, energy and the environment: a review of issues, evidence and policy. Urban Studies, 33(1): 7-35
(2) Line 54-55: Why the impact of urban form on carbon emission has not yet been systematically and accurately quantified? Did the previous studies ignore this part?
(3) Section 2.1: Some other similar terms, such as "urban configuration" and "spatial pattern", etc. could also be taken into consideration.
(4) I wonder why the authors started the bibliometric analysis from 2002?
(5) In the Discussion Section, it is suggested that an important part of urban form, three-dimensional urban structure, should also be discussed (see below for example).
Investigating the impacts of three-dimensional spatial structures on CO2 emissions at the urban scale. Sci. Total Environ. 2021, 762: 143096
Analyzing the impact of three-dimensional building structure on CO2 emissions based on random forest regression. Energy, 2021, 236: 121502
(6) It would be better if this manuscript could further summarize the methods (e.g., regression, correlation) used for analyzing the relationships between urban form and carbon emissions.
Author Response

(The authors gave the same response as above.)

Round 2
Reviewer 1 Report
I have no objection to the current version of the manuscript. The introduced changes improved the structure and substantive value of the article. I recommend publishing it.
Reviewer 3 Report
Thank you for considering my comments and suggestions. The revised manuscript is now ready for publication.